# Academic and Corporate Vehicle Electrification Research

**Hans Pohl [1],\* and Magnus Karlström [2]**

1   Lindholmen Science Park, P.O. Box 8077, 402 78 Gothenburg, Sweden
2   Department of Energy and Environment, Chalmers University of Technology, 412 96 Gothenburg, Sweden
\*   Correspondence: hans.pohl@lindholmen.se; Tel.: +46-708402740

**Abstract:** We developed and used methodology to analyze scientific publications in Scopus relating to vehicle electrification and associated key enabling technologies: batteries, fuel cells and electric machines with power electronics. The global research landscape was mapped, and an analysis of the 16 most active countries was carried out. Vehicle electrification publications are rewarded with a high citation impact, and they include corporate actors to a great extent. China dominates in vehicle electrification research as well as in the enabling technologies, and China's position is set to become even more dominating. Battery research has grown rapidly with a high citation impact, whereas the volume of research for the other enabling technologies was more constant during 2017–2021. Automakers' research that has led to scientific publications was specifically studied. Ford Motor Company was the automaker with the highest number of vehicle electrification publications during 2017–2021. A large share of the automakers' publications was co-authored with academic actors, and such publications were rewarded with a higher citation impact than those without. However, the share of international co-publications among the automakers was meager. It is concluded that the analysis of vehicle electrification publications gives an overview of the rapidly developing field. Moreover, the analysis of automakers' involvement in such research is one way of obtaining one perspective on their strategies and priorities.

**Keywords:** BEV (battery electric vehicle); EV (electric vehicle); fuel cell vehicle; HEV (hybrid electric vehicle); research publications; scientometrics





## 1. Introduction

Electrified vehicles, including batteries and fuel cells, are on the market, and the volumes of electric cars, trucks and buses are growing. Nevertheless, new solutions and knowledge are still needed to broaden the market and reduce the environmental impact. This paper used scientific publications to study knowledge development related to vehicle electrification.

Quantitative studies of vehicle electrification using publications typically focus rather narrowly on electric vehicles and specific geographic areas. However, publications explicitly mentioning "electric vehicles" or similar terms constitute only a tiny part of the total research body relevant to vehicle electrification. For example, very few articles within battery research mention specific applications as they are on a more generic level.

The purpose is to better understand vehicle electrification research. This is achieved through mapping of the global vehicle electrification research landscape using scientific publications. To manage this mapping, new publication analysis methodology had to be developed. This mapping is relevant to understand how different countries are involved and how research in this field differs from other fields. A specific focus of the study is on automakers and their collaborations with academia. The results can be used to guide research policies, and the methodology used can inspire other researchers within the field of scientometrics.

This article develops new methodology to capture a broader range of publications relevant to vehicle electrification than in previous studies. In addition to the study of

"vehicle electrification" publications, three enabling technologies are added and studied separately. They are batteries, fuel cells and electric machines and power electronics. Another contribution of the study is the analysis of a selection of automakers and how they collaborate with academic actors.

Several questions are addressed by analyzing Scopus publications from 1996 until and including 2021 about electrified vehicles and key enabling technologies. Which countries and automakers are most active? What is their citation impact? How are they collaborating?

Following a brief review of previously published studies of vehicle electrification, the methodology and data are outlined. Thereafter, results are presented and divided into two parts, one giving national data and one describing how automakers are involved in academic publications. Finally, discussions and conclusions follow.

## 2. Previous Literature

From the previously published studies of vehicle electrification, typically, a few keywords or classifications provided by the database supplier can be used to identify relevant publications [1–7]. They focus on electric vehicles, which means publications explicitly mentioning both electrification and an automotive application, in some cases using search terms such as "electric vehicle" or similar [cf. 2 and 5]. Some studies try to gain a comprehensive view of electric vehicle technologies and understand the emphasis of current research. Web of Science dominates as the source of data. Among the results found are that vehicle electrification publications are rapidly increasing in volume, and China is gaining a dominating role in producing such publications.

There are also published studies addressing the enabling technologies. In [8], Li-ion battery research is studied using a selection of search terms. Rather detailed results indicating the most productive researchers are presented. Based on the most frequent keywords, hotspots are suggested. Another study of battery-related publications focuses on how the European Battery 2030+ initiative is positioned. It uses relatively advanced clustering methodology to identify and analyze relevant publications. Among the results, the United States and Canada tend to lead in citation impact, whereas China dominates in publication volumes [9]. An even more specific battery study addresses battery states of charge and health [10]. It uses both Web of Science and Scopus data to identify top authors and institutions as well as some trends. Additionally, one rather specific, published study focuses on wireless charging [11]. It uses a small selection of search terms combined with the manual scrutiny of all resulting publications to obtain a set of publications for further analysis.

The study of the previous literature highlights some critical aspects. First, it is important to have competence in vehicle electrification and scientometrics. Second, a clear description of the search strategy employed is critical for interpreting the results. Third, a narrow search extracting only "electric vehicle" publications might miss important developments in the technologies enabling vehicle electrification. Finally, it could also be noted that in most cases, types of data other than publication data are not used.

## 3. Methodology and Data

SciVal, a tool provided by Elsevier, was used to extract and analyze Scopus publications. Scopus is a publication database with broader coverage than Web of Science [12]. Data were extracted in December 2022, meaning that the last almost complete year was 2021. Elsevier considers the publication data for a year sufficiently complete in May of the following year. However, this does not apply to citation indicators, which are based on accumulating data over a more extended period of time.

The identification of publications was based on queries. Queries use search terms which can be combined with AND, OR, and AND NOT. In addition, entities such as countries, institutions, and authors were used. The search strategy is illustrated in Figure 1. Vehicle electrification publications, from now on called XEV publications, are the sum of



"Vehicle electrification" and the overlapping areas joining the enabling technologies and the automotive domain (Figure 1 is not to scale).

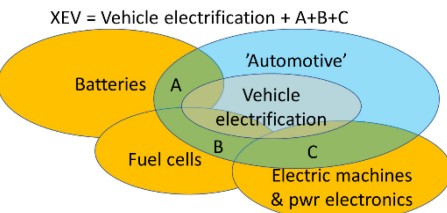

**Figure 1.** Search query for vehicle electrification publications (XEV).

One challenge is that publications in enabling technologies clearly relevant to vehicle electrification seldom mention automotive applications in the title, keywords, or abstract. Only a few percent of battery publications mention automotive applications. One of Sweden's most productive battery researchers estimated that 50% of her publications addressed issues relevant to automotive applications. However, less than 3% were identified by the query for XEV publications. The strategy to manage this challenge was to include separate studies of publications on enabling technologies. The queries were designed in an iterative process; see Figure 2.

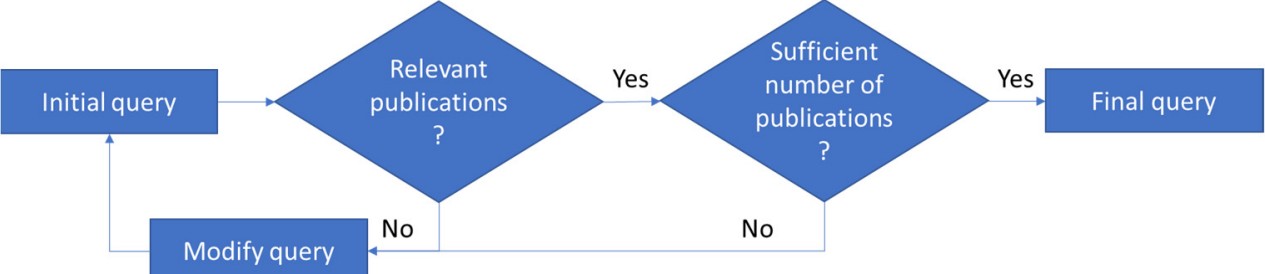

**Figure 2.** Query design process.

The manual scrutiny of 100 randomly selected publications' titles, abstracts, and journal names was used to determine if the extracted publications were within the field of vehicle electrification. The next and more difficult step was to determine if all relevant publications were included. This was addressed using publication lists from researchers in the field as well as all publications from the Swedish Electromobility Centre. Further details about the methodology are given in [13].

Even though we iterated the queries many times to maximize the number of correctly identified publications and minimize the number of falsely included ones, the resulting lists are incomplete. The number of relevant publications is likely higher. Our study underestimates the total. Another known and probably unavoidable problem is a bias towards more applied research. For obvious reasons, blue sky research seldom mentions potential applications, and they are therefore not captured by the queries. This is true for the XEV query as well as for the queries for the enabling technologies.

Preliminary results and the methodology were discussed with researchers and other actors working with vehicle electrification. Approximately 40 people were consulted, among them being the R&D managers at the three large automakers in Sweden: Volvo Cars, AB Volvo, and Scania. Thereafter, modifications were made.

## 4. Results

### 4.1. Overall Description of Data

First, word clouds for 2012 and 2021 were compared. They were based on Elsevier's Fingerprint engine, which uses the title, abstract, and keywords to generate key phrases. The larger the font, the higher the relevance. To some extent, the word clouds mirror the

query used to extract the publications. However, some search terms are clearly represented in the word clouds, whereas others are not. The green word cloud in Figure 3 is based on 12,809 XEV publications from 2021. The blue word cloud in Figure 4 uses 5410 XEV publications from 2012.

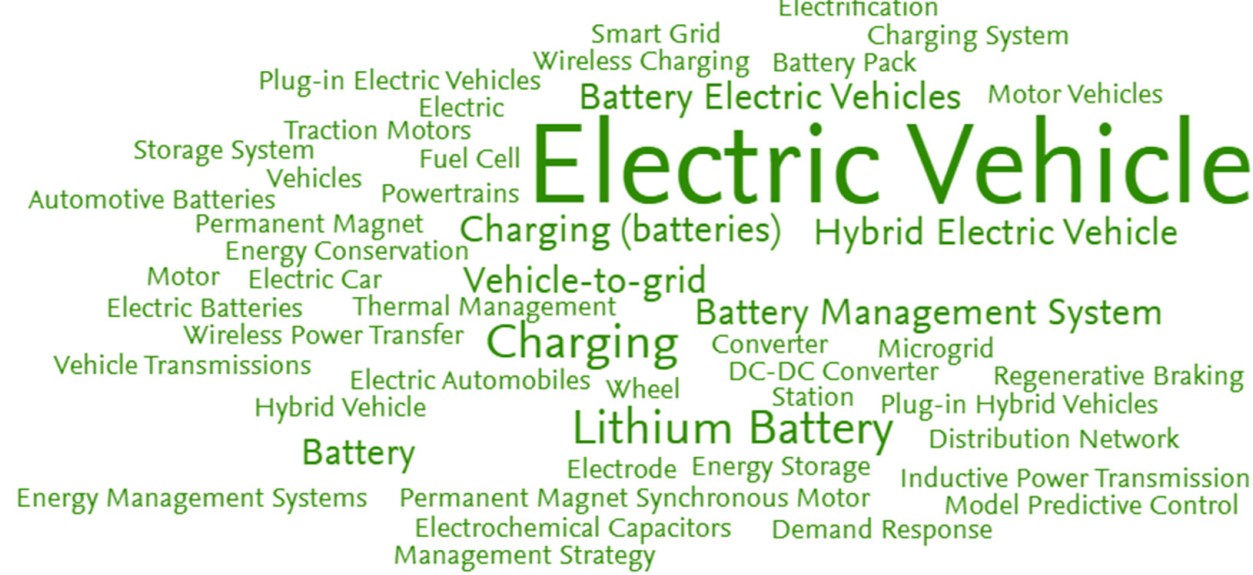

**Figure 3.** Key phrases based on XEV publications in 2021.

**Figure 4.** Key phrases based on XEV publications in 2012.

During this period, the focus of research changed. In 2012, different types of hybrid electric vehicles constituted substantial portions of research. In 2021, hybrid vehicles were less in focus. Key phrases related to charging indicate that this type of research produced approximately the same share of publications in both years. Fuel cells were mentioned in 2012, albeit with just one key phrase. In 2021, there was no fuel-cell-related key phrase.

Given the design of XEV publications as a combination of vehicle electrification publications and publications in the enabling technologies mentioning automotive applications, it is interesting to study to what extent the enabling technology publications mention electric vehicle applications. In Figure 5, the darker blue areas represent the publications

identified as both XEV and within one or several enabling technologies. For example, out of 24,228 battery publications, 12% or 2797 were also XEV publications.

**Figure 5.** Overlap between XEV and enabling technology publications (2021).

As explained in the Methodology section, Figure 5 should not be interpreted as an indication that most of the enabling technology research is not relevant to vehicle electrification. Rather, it shows that enabling technology research is often carried out on a more basic level, where the potential applications are not relevant to mention. Moreover, it should be noted that the query to identify enabling technology publications is biased towards more applied research.

### 4.2. National Perspectives

The volumes of vehicle electrification (XEV) publications increased far more than the overall increase in publications. Figure 6 shows the annual production of the 16 countries with the highest numbers of XEV publications. In 2011, China overtook the United States in XEV publication volume, and in 2021, China also passed India.

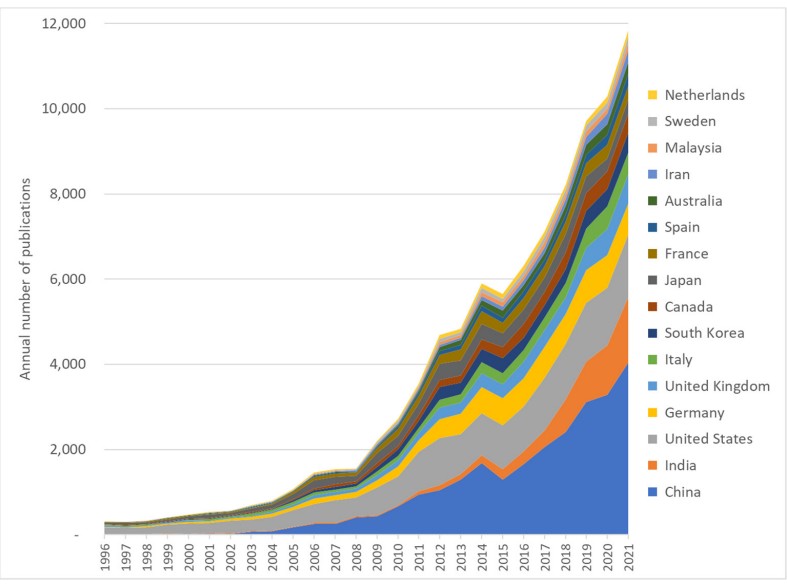

**Figure 6.** Annual vehicle electrification publication volumes.

Table 1 features some indicators characterizing the XEV publications per country. Data for all publications in each country are also given to allow for comparisons. In line with Figure 3, China's publication volume was the largest, with almost 15,000 publications over the five-year period 2017–2021, more than twice the volume of XEV publications produced in the second largest country, the United States. On the other hand, India was only larger than the United States in 2021, which means that during the period, it was the third largest country.

**Table 1.** Key indicators for XEV publications in 2017–2021.

| | Publication Volume XEV 2017–2021 | Annual Publication Volume Growth 2017–2021 (CAGR) | | Citation Impact (FWCI) | | Share of Academic-Corporate Co-Publications | | International Co-Publications (FWIS) | |
|---|---|---|---|---|---|---|---|---|---|
| | | XEV | Nt'l | XEV | Nt'l | XEV | Nt'l | XEV | Nt'l |
| Australia | 1110 | 10.3% | 3.3% | 2.25 | 1.61 | 3.1% | 3.3% | 1.59 | 1.39 |
| Canada | 1988 | 6.2% | 3.0% | 2.08 | 1.47 | 6.6% | 4.3% | 1.45 | 1.32 |
| China | 14,905 | 14.4% | 10.2% | 1.35 | 1.08 | 6.3% | 2.8% | 0.59 | 0.54 |
| France | 1531 | 3.2% | 0.4% | 1.41 | 1.30 | 12.9% | 6.2% | 1.57 | 1.35 |
| Germany | 3706 | 0.7% | 1.9% | 1.40 | 1.33 | 17.6% | 6.5% | 0.77 | 1.21 |
| India | 4815 | 28.6% | 8.7% | 1.12 | 0.93 | 2.2% | 1.2% | 0.43 | 0.47 |
| Iran | 997 | 18.3% | 6.7% | 1.99 | 1.09 | 0.7% | 0.8% | 1.20 | 0.67 |
| Italy | 2132 | 12.2% | 5.5% | 1.78 | 1.44 | 9.5% | 4.1% | 1.06 | 1.13 |
| Japan | 1774 | −3.3% | 1.4% | 1.09 | 0.93 | 13.2% | 6.6% | 0.69 | 0.72 |
| Malaysia | 643 | 5.2% | 4.9% | 1.52 | 1.08 | 0.9% | 1.6% | 1.25 | 1.16 |
| Netherlands | 645 | 4.8% | 2.9% | 1.85 | 1.73 | 11.9% | 7.3% | 1.28 | 1.52 |
| South Korea | 1887 | 10.3% | 3.9% | 1.28 | 1.08 | 9.4% | 4.5% | 0.57 | 0.70 |
| Spain | 1030 | 14.5% | 5.3% | 1.48 | 1.28 | 9.9% | 3.9% | 1.31 | 1.17 |
| Sweden | 681 | 6.8% | 2.5% | 1.89 | 1.62 | 11.9% | 7.4% | 1.25 | 1.56 |
| United Kingdom | 2614 | 14.0% | 1.5% | 2.15 | 1.56 | 11.6% | 5.5% | 1.55 | 1.38 |
| United States | 6722 | 3.1% | 0.7% | 2.08 | 1.37 | 8.2% | 4.7% | 1.08 | 0.87 |
| World | 50,262 | 10.5% | 4.1% | 1.37 | 1.00 | 6.4% | 2.7% | 0.53 | 1.00 |

Green color indicates maximum value in each column.

In the third and fourth columns, the publication volume growth from 2017 to 2021 was based on a linear regression approximation of the data points for each country and year, followed by the compounded annual growth rate calculation. Again, India showed very rapid growth. For most countries, XEV publication volumes grew more rapidly than the national average, which was also confirmed on the world level, with 10.5% growth for XEV versus 4.1% for all publications.

The field-weighted citation impact (FWCI) is a proxy for publication quality. It is a normalized indicator comparing the number of citations that a publication receives with the citation volumes for other publications in the same scientific field, published in the same year, in the same type of publication. The global average is 1. Almost all countries have higher FWCIs for XEV publications than for average publications. To some extent, a high FWCI confirms that the field is "hot" and rapidly growing.

The share of academic-corporate co-publications reflects academic-corporate collaboration. XEV publications involve corporate actors to a higher degree than average publications. In several countries, the shares are very high and are investigated in more detail below.

Finally, in Table 1, another normalized indicator based on international co-publications with a global average of 1 (see [14]) shows no clear pattern regarding how international the vehicle electrification research is. However, a comparison of Germany and France

reveals large differences. Whereas Germany has a much lower share of international co-publications in the XEV area than nationally, France has a slightly higher score.

In relative terms, which countries are most active in XEV research? In Table 2, three relative indicators are given, comparing the XEV publication volumes with automotive, national, and global volumes of publications.

**Table 2.** Relative XEV publications (2017–2021).

| | XEV Share of Automotive | XEV Share of National | Share of XEV World |
|---|---|---|---|
| Australia | 31% | 0.19% | 2.2% |
| Canada | 41% | 0.33% | 4.0% |
| China | 43% | 0.42% | 29.7% |
| France | 33% | 0.24% | 3.0% |
| Germany | 29% | 0.38% | 7.4% |
| India | 32% | 0.48% | 9.6% |
| Iran | 41% | 0.30% | 2.0% |
| Italy | 33% | 0.31% | 4.2% |
| Japan | 34% | 0.25% | 3.5% |
| Malaysia | 23% | 0.34% | 1.3% |
| Netherlands | 31% | 0.19% | 1.3% |
| South Korea | 34% | 0.41% | 3.8% |
| Spain | 30% | 0.19% | 2.0% |
| Sweden | 26% | 0.29% | 1.4% |
| United Kingdom | 35% | 0.23% | 5.2% |
| United States | 31% | 0.19% | 13.4% |

Green color indicates maximum value in each column.

China has the highest share of XEV publications when related to the total volume of automotive publications. This could, to some extent, depend on China's relatively late involvement in the production of road vehicles. The internal combustion engine has probably created large communities of researchers in countries with long traditions in vehicle development and production. The capacity to develop the internal combustion engine is still relevant, even though its role as a key performance ingredient is rapidly decreasing. Another potential explanation for China's high share in this comparison might be that China has had a relatively strong policy promoting vehicle electrification.

India leads when the XEV publication volume is compared with all publications made with Indian researchers as (co-)authors. This means that the share of automotive research in India is rather large. It is surprising that several of the traditional countries with large automotive industries do not appear to have a corresponding volume of research. However, a part of the explanation could also be that the query used in this study has bias towards more applied research. As described in the Methodology section using the case of batteries, most of the battery publications do not mention automotive applications, even though it might be relevant.

The final column once again illustrates China's dominance in XEV publications. As the table is based on an average of five years, and China's research is still growing more rapidly than the world average, the share in 2021 is even larger than 29.7%.

As described in the Methodology section, a separate analysis of enabling technologies was also carried out. Given the vast range of applications for these technologies, the volume of research is not directly linked to the capability to develop more advanced electrified vehicles. However, it is probable that a large share of the researchers in these fields can contribute.

### 4.2.1. Battery Publications

Batteries are needed for all types of electrified vehicles. Many chemistries compete, and the processes within the battery cells are challenging to understand. The volumes of

scientific publications increased rapidly during 2017–2021, which is in line with the trend from previous years (see Table 3).

**Table 3.** Key indicators for battery publications.

| | Publication Volume Batteries 2017–2021 | Publication Volume Growth 2017–2021 (CAGR) | | Batteries Share of Automo-tive | Batteries Share of National | Share of Batteries World |
|---|---|---|---|---|---|---|
| | | Batteries | National | | | |
| Australia | 3829 | 12.2% | 3.3% | 108% | 0.67% | 3.6% |
| Canada | 2859 | 9.8% | 3.0% | 58% | 0.47% | 2.7% |
| China | 60,614 | 13.7% | 10.2% | 176% | 1.73% | 56.5% |
| France | 2572 | 4.0% | 0.4% | 55% | 0.40% | 2.4% |
| Germany | 6093 | 6.2% | 1.9% | 48% | 0.63% | 5.7% |
| India | 8228 | 15.9% | 8.7% | 55% | 0.82% | 7.7% |
| Iran | 1427 | 13.4% | 6.7% | 59% | 0.43% | 1.3% |
| Italy | 1737 | 9.0% | 5.5% | 27% | 0.25% | 1.6% |
| Japan | 4483 | 5.0% | 1.4% | 86% | 0.64% | 4.2% |
| Malaysia | 1215 | 7.3% | 4.9% | 43% | 0.64% | 1.1% |
| Netherlands | 548 | 6.6% | 2.9% | 26% | 0.16% | 0.5% |
| South Korea | 8041 | 7.1% | 3.9% | 146% | 1.74% | 7.5% |
| Spain | 1941 | 10.4% | 5.3% | 57% | 0.36% | 1.8% |
| Sweden | 1101 | 11.3% | 2.5% | 43% | 0.47% | 1.0% |
| United Kingdom | 3850 | 13.3% | 1.5% | 51% | 0.34% | 3.6% |
| United States | 15,814 | 2.3% | 0.7% | 73% | 0.44% | 14.7% |
| World | 107,282 | 8.0% | 4.1% | | | |

Green color indicates maximum value in each column.

China dominates, with 56.5% of the publications and rapid growth, which indicates that the share will soon be even higher. South Korea and Japan, which historically have been dominating battery development and production in the world, exhibit rather low shares of battery publications and a less rapid increase than the world average. Still, battery research represents the largest share of all publications in South Korea.

The fact that some countries have higher volumes of battery than automotive publications highlights the need for a separate analysis of research relating to the enabling technologies. Even though far from all battery-related publications represent knowledge of relevance for automotive applications, a large share probably does. A narrow study of the small share of "automotive battery publications" might miss important aspects.

The citation impact for battery publications is very high, and this is true for the world as well as for almost all countries. On average, a battery publication is cited approximately twice as much as an average publication. Battery publications with Australian participation enjoy the highest FWCI (2.98), followed by the United States (2.56) and Canada (2.52). Chinese battery research is cited more than the average battery publication, with a FWCI of 2.16.

### 4.2.2. Fuel Cell Publications

Fuel cells for automotive applications are predominantly of the polymer electrolyte (PEM) type, which means they operate at relatively low temperatures. Therefore, the query extracting fuel cell publications included all types. As highlighted in Table 4, fuel cell research is, when measured by publication volumes, approximately half the size of battery research. Moreover, the volume is increasing at a slightly lower rate than the overall publication volume, which means that the share is decreasing.

**Table 4.** Key indicators for fuel cell publications.

| | Publication Volume Fuel Cells 2017–2021 | Publication Volume Growth 2017–2021 (CAGR) | | Fuel Cells Share of Automotive | Fuel Cells Share of National | Share of Fuel Cells World |
|---|---|---|---|---|---|---|
| | | Fuel Cells | National | | | |
| Australia | 1049 | 0.9% | 3.3% | 30% | 0.18% | 2.2% |
| Canada | 1538 | 0.2% | 3.0% | 31% | 0.26% | 3.2% |
| China | 16,294 | 9.7% | 10.2% | 47% | 0.46% | 34.1% |
| France | 1707 | 0.3% | 0.4% | 36% | 0.27% | 3.6% |
| Germany | 2517 | −1.7% | 1.9% | 20% | 0.26% | 5.3% |
| India | 4451 | 8.7% | 8.7% | 30% | 0.44% | 9.3% |
| Iran | 1785 | 8.7% | 6.7% | 74% | 0.54% | 3.7% |
| Italy | 1580 | −2.4% | 5.5% | 24% | 0.23% | 3.3% |
| Japan | 2741 | −3.5% | 1.4% | 53% | 0.39% | 5.7% |
| Malaysia | 1074 | 5.9% | 4.9% | 38% | 0.57% | 2.2% |
| Netherlands | 376 | −1.5% | 2.9% | 18% | 0.11% | 0.8% |
| South Korea | 2847 | 4.6% | 3.9% | 52% | 0.61% | 6.0% |
| Spain | 1063 | 0.9% | 5.3% | 31% | 0.20% | 2.2% |
| Sweden | 543 | 0.0% | 2.5% | 21% | 0.23% | 1.1% |
| United Kingdom | 2093 | 3.5% | 1.5% | 28% | 0.18% | 4.4% |
| United States | 5667 | −2.5% | 0.7% | 26% | 0.16% | 11.9% |
| World | 47,776 | 3.9% | 4.1% | | | |

Green color indicates maximum value in each column.

China also dominates in fuel cell research, with approximately one third of all publications in the world and the fastest increase over the period of 2017–2021. On the other hand, India also had a high growth rate and has, after the United States, the third largest share in the world. As for batteries, fuel cell research in South Korea has the highest share of the overall production in the country.

The low or even decreasing volumes of fuel cell research in several countries that have been very active in the development of such vehicles for at least 25 years, among them being Japan, South Korea, and Germany, could either signal that the technology is not very high on the agenda any longer or that the need for more research is not so strong. As hydrogen and fuel cells have reappeared as priorities in national and international policies, not least for heavier vehicles, it is probable that the latter is the case. If so, it could indicate that the technologies needed are available and that it is now mainly a matter of engineering to make them competitive.

4.2.3. Electric Machines and Power Electronics

Finally, the development of research in electric machines and power electronics is the most general field, which means that it does not link so strongly to vehicle electrification. The volume of publications is on par with the battery field, but the growth rate is clearly lower; see Table 5.

India is the second largest producer of publications in this field after China, which makes more than one-quarter of all publications and has the highest growth rate. As for fuel cells, countries have shown both positive and negative trends over the past five years. Iran was the only country with a higher number of publications in this field than automotive ones. This is explained in part by the fact that Iran has a relatively limited total volume of automotive publications.

**Table 5.** Key indicators for electric machines and power electronics publications.

| | Publication Volume Electric Machines and Power Electronics 2017–2021 | Publication Volume Growth 2017–2021 (CAGR) | | Machines Share of Automotive | Machines Share of National | Share of Machines World |
|---|---|---|---|---|---|---|
| | | Machines | National | | | |
| Australia | 1640 | 5.0% | 3.3% | 46% | 0.29% | 1.7% |
| Canada | 2434 | 0.1% | 3.0% | 50% | 0.40% | 2.6% |
| China | 25,297 | 8.3% | 10.2% | 73% | 0.72% | 26.9% |
| France | 2813 | −3.1% | 0.4% | 60% | 0.44% | 3.0% |
| Germany | 4765 | 0.5% | 1.9% | 37% | 0.49% | 5.1% |
| India | 11,571 | 3.0% | 8.7% | 77% | 1.16% | 12.3% |
| Iran | 2614 | 7.2% | 6.7% | 108% | 0.79% | 2.8% |
| Italy | 3068 | 2.8% | 5.5% | 47% | 0.45% | 3.3% |
| Japan | 3615 | −6.1% | 1.4% | 69% | 0.52% | 3.8% |
| Malaysia | 1174 | −5.3% | 4.9% | 42% | 0.62% | 1.2% |
| Netherlands | 603 | 6.5% | 2.9% | 29% | 0.18% | 0.6% |
| South Korea | 3149 | −0.5% | 3.9% | 57% | 0.68% | 3.3% |
| Spain | 2032 | 5.4% | 5.3% | 60% | 0.38% | 2.2% |
| Sweden | 643 | −0.2% | 2.5% | 25% | 0.28% | 0.7% |
| United Kingdom | 3990 | 1.1% | 1.5% | 53% | 0.35% | 4.2% |
| United States | 9496 | −0.8% | 0.7% | 44% | 0.26% | 10.1% |
| World | 94,158 | 2.2% | 4.1% | | | |

Green color indicates maximum value in each column.

*4.3. Automakers' Involvement in Research*

When looking at scientific publications, a less common perspective is how the corporate actors participate. Data for a selection of automakers representing different countries and vehicle types and two Tier 1 suppliers are presented in Table 6. Only automakers with substantial publication volumes were included in the analysis, which, among others, means that companies only making electric vehicles such as Tesla were not included. Robert Bosch has the largest total volume of scientific publications, whereas Ford Motor Company has the highest number of publications relating to vehicle electrification, albeit with a negative trend.

Toyota had the highest share of electrification publications during the period. In terms of vehicle output, Toyota has been and is the largest or second largest automaker. It made a clear commitment to vehicle electrification by developing and launching the hybrid electric Toyota Prius in 1997. Since then, it has introduced many hybrid models and has also been a pioneer in the market introduction of fuel cell vehicles. It is therefore not so surprising that Toyota has the largest share of XEV publications.

Compared to the analysis in Section 4.1, this subsection includes all publications where the automotive actor has an affiliation as it is very probable that their research is related to vehicles. However, it should also be noted that the volumes of XEV publications per automaker are relatively small, which calls for some caution when interpreting the results.

The research managers at the Swedish OEMs argued that the volumes should be related to the size of the company. When the volumes were related to the turnover or the official data for investments in research and development, the smaller companies (including those in Sweden) indicated the highest relative research intensity.

Another perspective on automakers' publications is to what extent they are co-authored with academic actors. The definition of an academic–corporate co-publication is that it has at least two co-authors and at least one academic and one corporate affiliation.

This means that double or multiple affiliations including academic and corporate actors also qualify, as long as two authors are involved. In Figure 7, the FWCI is indicated for publications with and without such collaboration.

**Table 6.** Electrification publications with corporate (co-)authors.

| Scopus Publications 2017–2021 | All | | XEV | | Share |
|---|---|---|---|---|---|
| | Volume | Trend | Volume | Trend | Electrification |
| BMW Group | 1333 | 3.1% | 136 | 14.0% | 10.2% |
| Daimler AG | 1163 | 0.1% | 189 | 8.0% | 16.3% |
| FAW Group Corporation | 575 | 3.8% | 48 | 8.8% | 8.3% |
| Ford Motor Company | 2187 | −8.6% | 310 | −12.8% | 14.2% |
| General Motors | 1708 | −7.9% | 284 | −7.7% | 16.6% |
| Honda Motor Co., Ltd. | 920 | 4.3% | 87 | −4.1% | 9.5% |
| Hyundai Motor Group | 1316 | 7.1% | 120 | 7.3% | 9.1% |
| Scania AB | 203 | −5.6% | 24 | 0.0% | 11.8% |
| Toyota Motor | 1191 | −3.4% | 242 | −11.4% | 20.3% |
| Volkswagen AG | 1732 | 2.4% | 184 | −0.2% | 10.6% |
| Volvo Car Corporation | 568 | −0.1% | 42 | 1.9% | 7.4% |
| Volvo Group | 462 | −2.0% | 36 | 17.8% | 7.8% |
| DENSO Corporation | 716 | −13.1% | 96 | −12.1% | 13.4% |
| Robert Bosch GmbH | 2497 | 2.2% | 258 | −8.9% | 10.3% |

Green color indicates maximum value in each column.

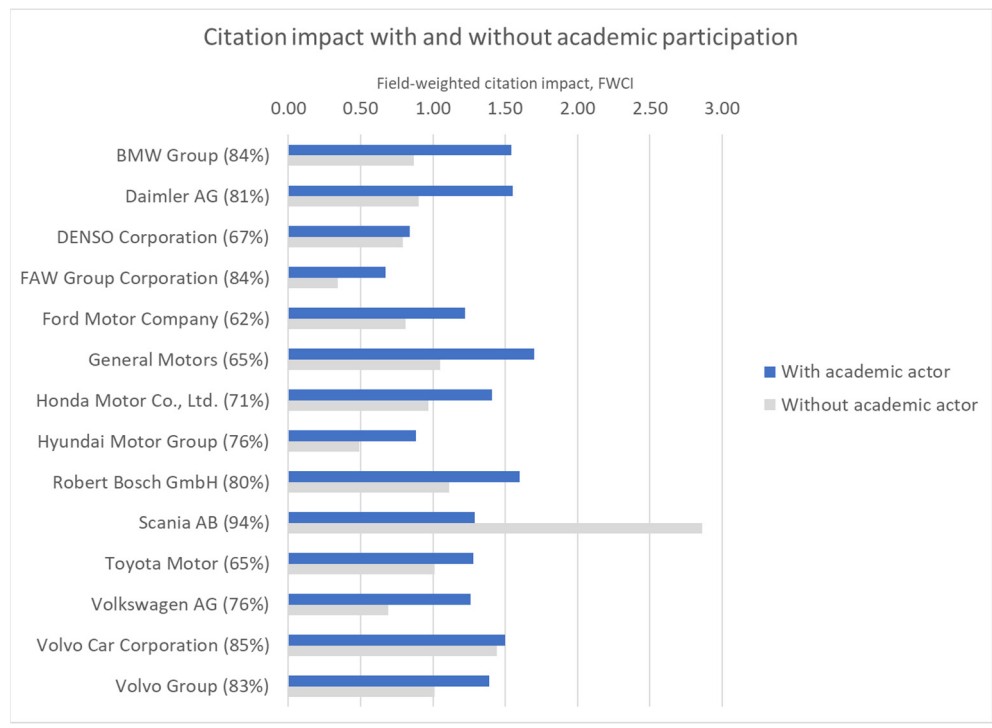

**Figure 7.** Automaker publications with and without academic actors in 2017–2021.

Directly behind each automaker's name, the percentage indicates how large the shares of publications including academic co-authors are. For example, it could be noted that Scania almost exclusively publishes with academic participation and that the automaker with the highest share of in-house publications is Ford Motor Company. The main message in Figure 7 is that collaboration with academic actors leads to a higher citation impact. Scania is an exception, but it should be noted that the number of publications without academic participation is minimal (12 publications).

Sweden's FWCI for the period was 1.62 (see Table 1). Publications involving the three automakers in Sweden had clearly lower FWCIs. In discussion with the R&D managers from the automakers, the central hypothesis was that a large share of the publications was made by junior researchers, not least PhD candidates. This could have led to a lower citation impact.

In the highly international automotive industry, it could be expected that research collaboration is also international. The main trend in research is towards more international co-publications, and academic–corporate co-publications are no exception; cf. [15]. However, the analysis shows that publications involving the automakers are largely national, and the academic partner is often even regional, which means the partner is in the same city as the main research facility; see Table 7. General Motors has more international than national publications; otherwise, the share of international co-spublications is much lower than those in the home country of the automaker.

**Table 7.** International partners among the top ten collaborations.

| Scopus Publications 2017–2021 | Top 3 Partner Institutions | | | | | | Among Top 10 Partners | |
|---|---|---|---|---|---|---|---|---|
| Automotive Company | Name | No. of Co-Pubs. | Name | No. of Co-Pubs. | Name | No. of Co-Pubs. | No. of Int'l | Share of Publ. with Int'l |
| BMW Group | Technical University of Munich | 378 | Karlsruhe Institute of Technology | 63 | Universität der Bundeswehr München | 48 | 2 | 6.4% |
| Daimler AG | University of Stuttgart | 128 | Ulm University | 127 | Karlsruhe Institute of Technology | 97 | 0 | 0.0% |
| FAW Group Corporation | Jilin University | 291 | Volkswagen AG | 23 | Harbin Institute of Technology | 22 | 1 | 5.3% |
| Ford Motor Company | University of Michigan, Ann Arbor | 174 | Ohio State University | 96 | Michigan State University | 77 | 2 | 12.4% |
| General Motors | Université du Québec à Chicoutimi | 67 | University of Waterloo | 66 | University of Michigan, Ann Arbor | 60 | 5 | 56.2% |
| Honda Motor Co., Ltd. | Tokyo Institute of Technology | 47 | Kyoto University | 35 | Technische Universität Darmstadt | 31 | 3 | 29.9% |
| Hyundai Motor Group | Seoul National University | 156 | Hanyang University | 140 | Korea Advanced Institute of Science and Technology | 70 | 0 | 0.0% |
| Scania AB | KTH Royal Institute of Technology | 111 | Uppsala University | 21 | Chalmers University of Technology | 15 | 0 | 0.0% |
| Toyota Motor | Toyota Central R&D Labs., Inc. | 148 | The University of Tokyo | 92 | Nagoya University | 72 | 0 | 0.0% |
| Volkswagen AG | Technical University of Braunschweig | 157 | Technical University of Munich | 123 | Karlsruhe Institute of Technology | 84 | 1 | 7.8% |
| Volvo Car Corporation | Chalmers University of Technology | 315 | KTH Royal Institute of Technology | 42 | University of Gothenburg | 37 | 0 | 0.0% |
| Volvo Group | Chalmers University of Technology | 146 | Eicher Motors Limited | 78 | Mälardalen University | 37 | 3 | 27.9% |
| DENSO Corporation | Nagoya University | 68 | Toyota Motor | 37 | Tohoku University | 35 | 0 | 0.0% |
| Robert Bosch GmbH | University of Stuttgart | 216 | Karlsruhe Institute of Technology | 190 | Ulm University | 99 | 0 | 0.0% |

As Table 7 indicates, half of the automotive companies have no international partner in their list of the ten largest research collaborations. Among those with at least one international partner, it tends to be at the bottom of the list, thus constituting a low share of the total. The location of the largest partner institution shows, in many cases, where the main research facility is based.

Another rule of thumb is that international co-publications are more cited than national co-publications. In Figure 8, the FWCIs for national and international publications are indicated. In this regard, the automotive industry is in line with the norm, and international co-publications are typically cited more often. The numbers behind the names of the automakers are the shares of national co-publications in the period.

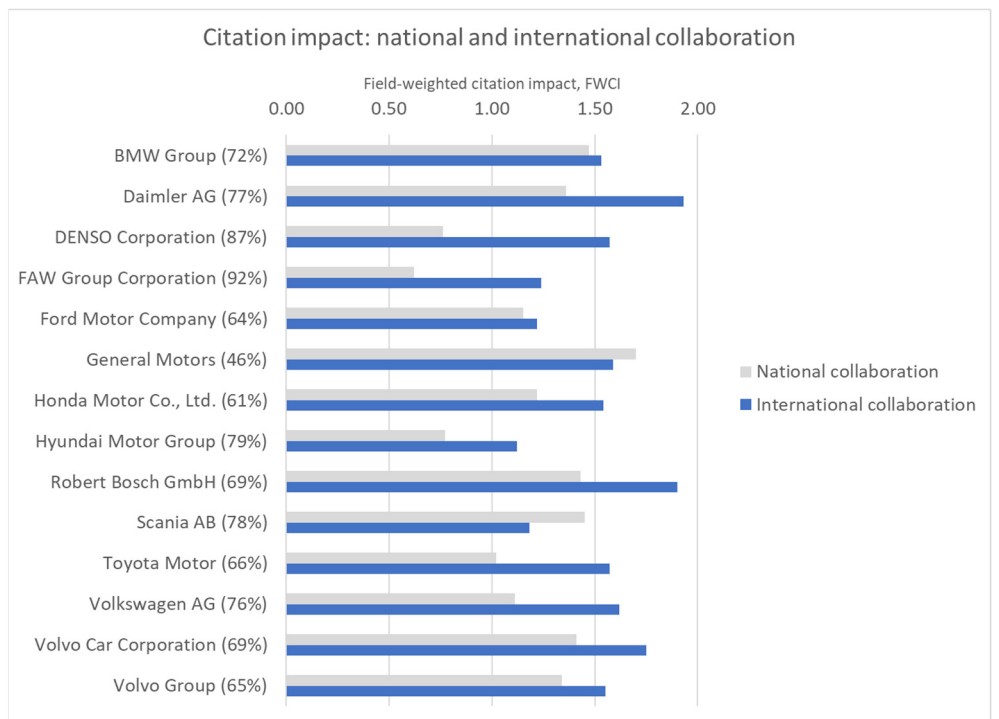

**Figure 8.** National and international co-publications in 2017–2021.

## 5. Discussion

The method and data differ from previous studies. The Scopus database is broader and includes more publications in the softer sciences than Web of Science, which was used in most other studies. Another advantage of the study is that the development of the search queries is clearly presented, contributing to increased transparency. Even though it is almost impossible to tell how large a share of the publications within the enabling technologies is relevant for vehicle electrification, the inclusion of these technologies and publications allows for a better understanding of how the field is developing. Several indicators, such as the citation impact and the share of academic–corporate publications, are included, contributing to perspectives on how various countries approach vehicle electrification. However, the broad scope of the study is also a disadvantage, as many questions remain to be studied more in detail.

What do the results indicate? Even though the approach is somewhat different, the study confirms the trends towards more publications in China and India, and a lower share in Western countries, as noted in previous studies. China is now the largest producer of XEV publications as well as publications in all three enabling technologies selected. It should also be noted that it is not only quantity; publications including Chinese authors are also cited. For example, battery research with Chinese participation had a high FWCI, meaning that these publications were cited more than twice the global average for all scientific publications during 2017–2021.

The automotive industry has been and is very international in almost all aspects. Therefore, it is surprising that the research is not so international. Germany's XEV publications had an FWIS of 0.77, which is lower than the average publication globally (FWIS = 1) and much lower than the average German publication (FWIS = 1.21). Why? Is it because automotive research is more applied than typical research or is it funded differently? This aspect clearly deserves closer study.

The enabling technologies have developed differently. For example, the volume of battery publications has increased dramatically, whereas the fuel cell publications have not increased at all. Battery publications also have a very high citation impact. China dominates all three enabling technologies, but especially batteries, accounting for more than 50% of the global output.

Whereas battery electric vehicles now have been on the market with growing volumes for a couple of years, fuel cell vehicles are at least one step behind. This could lead to a shift in focus for battery electric vehicles towards more applied questions and thus less research generating publications. Obviously, this is not the case. An attempt to explain the difference could be that batteries are needed in all types of electrified vehicles and used in many different ways. This, combined with the still quite high number of competing battery chemistries and the often very complicated chemical reactions that constitute the function of the batteries, could motivate a continued high volume of research. Fuel cells are less complicated, and therefore, the focus might be more on engineering issues.

The study of publications including automakers (and to some extent, two Tier 1 suppliers) revealed that a large share of the publications also included academic actors. This was expected. There appear to be differences between US and Japanese automakers, with relatively lower shares of academic–corporate co-publications and European and South Korean automakers with higher shares. A capacity to make scientific publications in-house could be considered a strength, but it could also signal that the traditions differ between countries and companies.

As expected, publications including academic actors as well as international co-publications were rewarded with a higher citation impact. However, the strong preference for national and even local collaborations with academic actors might signal that aspects other than scientific excellence are important when selecting the academic partners. Obviously, one such aspect is the potential for recruitment. It is probably much easier to recruit new employees locally than nationally or internationally.

Overall, this study indicates a very rapid development of vehicle electrification research with new countries emerging and China becoming more and more dominant. As the decarbonization of road transport is a global challenge, it is encouraging to see that substantial efforts are being invested. However, limited international collaboration is troublesome. Overall, XEV publications are internationally co-authored half as much as average publications. There is a clear risk that parallel research is being carried out, which is inefficient. For competition reasons, it might be challenging for automakers to collaborate extensively with international partners. Nevertheless, universities and research institutes should be encouraged to intensify their international collaborations. Another type of collaboration which could have an impact is between universities and the automotive industry.

Finally, a study of publications is just one of many approaches available to obtain perspectives on research. One advantage of this approach is the possibility to cover the whole world, with the possibility of zooming in on individual institutions or researchers, if needed. Our publication study was discussed with some actors in the research system, thus enhancing the quality and increasing the relevance. The deep dive in automaker's publications gives a perspective on their involvement in vehicle electrification research, which otherwise is challenging to obtain.

## 6. Conclusions

This paper puts forward methodology to study research relating to vehicle electrification using Scopus publications. The results cover 16 countries with the highest number of

vehicle electrification publications. Among them, China is by far the largest, and China's dominance increased during 2017–2021. This is true for publications directly relating to vehicle electrification as well as for the enabling technologies of batteries, fuel cells and electric motors with power electronics. In comparison to average national publications, vehicle electrification publications are rewarded with higher citation impacts, and they also include corporate actors to a greater extent. It is argued that international collaboration should be promoted. Among publications, including the automotive industry, Robert Bosch has the highest total volume of scientific publications, whereas Ford Motor Company has the highest number of vehicle electrification publications. Toyota had the highest share of vehicle electrification publications. Co-publications with academic actors are dominating and appear to be rewarded in terms of higher citation impacts. International research collaboration is limited for most automakers. Vehicle electrification publications offer a relatively straightforward method to obtain an overview of the rapidly developing field. Moreover, the study of automakers' involvement in such research is a possible way of obtaining one perspective on their strategies and priorities, which otherwise is rather challenging to obtain.

**Author Contributions:** Conceptualization, H.P. and M.K.; methodology, H.P. and M.K.; software, H.P.; validation, H.P. and M.K.; formal analysis, H.P.; investigation, H.P. and M.K.; resources, H.P.; data curation, H.P.; writing—original draft preparation, H.P. and M.K.; writing—review and editing, H.P.; visualization, H.P.; project administration, H.P. and M.K.; funding acquisition, H.P and M.K. All authors have read and agreed to the published version of the manuscript.

**Funding:** This research was funded by the Swedish Energy Agency, the Swedish Transport Administration and the Region Västra Götaland.

**Data Availability Statement:** The main data are from Scopus, which are accessible from Elsevier.

**Conflicts of Interest:** The authors declare no conflict of interest. The funders had no role in the design of the study; in the collection, analyses, or interpretation of data; in the writing of the manuscript; or in the decision to publish the results.

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
