# Peer review of "Academic and Corporate Vehicle Electrification Research"

_wevj, doi:10.3390/wevj14030071_

Round 1

Reviewer 1 Report

Notes to the Authors:

The article presents the results of research on electric vehicles by mapping the global research landscape, indicating which countries and institutions are most active and what is the impact of their research on development. This study attempts to assess the most important issues related to electric vehicles using bibliometric analysis. The topics concern the areas of both mechanical engineering and transport economics, which are important for researchers and fit into the subject of the journal and certainly deserve attention, in particular, it concerns issues related to current problems related to electromobility and technological progress, including vehicle electrification and related key assistive technologies; batteries, fuel cells and electric machines with power electronics.

However, the scientific value of the article is negligible. The research goals (there are no hypotheses) are not well defined, and their verification on the basis of the presented considerations is highly debatable. The very structure of the article is not fully transparent. As is the incorrect wording of the article name. The work is weakened by the lack of research input - the analysis of the publication is based on the SciVal method and the biometric analysis of publications in individual years, the lack of a description of the statistical method or the use of the SWOT method, which does not fully allow for the verification of goals, and even more so because there are no hypotheses. This pattern of behavior is not enough. All considerations are not based on the authors' research, but are only descriptive. The disadvantage is the lack of reference to the latest research and publications conducted by such manufacturers as: TOYOTA, MERCEDES, BMW and other scientists based on selected research methods and the complete omission of considerations regarding TCO economic solutions, residual value of the vehicle, costs of 1kWh, e.g. extensively described in the available literature item. There is no reference to one of the most important determinants of the development of analyzes and forecasts of demand and supply. Throughout the work, the authors incorrectly use the term electric vehicles or electromobility, because they also describe low-emission vehicles. In addition, there is no information on publications in the field of economic aspects, in particular costs, purchase price of the vehicle, operation, price per kWh - the presented considerations are insufficient. The work is most weakened by the lack of the authors' contribution in the form of an innovative approach based, among others, on based on statistical or algorithmic methods - presentation of your own model. The authors base their research solely on publicly available data, using a simple research scheme. The very verification of even incorrectly defined work goals on the basis of the adopted scheme of procedures used (without specifying constraints, output scenarios, formulas, algorithms, etc.) is highly debatable, and their scientific value is negligible, taking into account the methodological and logical errors in formulating the goals themselves and how to verify them.

Detailed notes:

1) The title of the article - does not correspond to the content of the article - it should be redrafted

2) Abstract - requires re-editing. In the current form, e.g. (selection of the research sample, assumptions, measurements and methods, dates of tests - what issues were analyzed, research period), key results and main conclusions. Contribution to the state of the art and practical application.

3) Keywords - need to be checked again and defined correctly.

4) The introduction requires redrafting - it does not contain the context  the description of the basics of electromobility development, demand and supply forecasts, energy costs, restrictions on EURO emission standards - technical progress in recent years, the most important legal acts, key factors affecting publications and research. The current content has not been quoted, the purpose of the research and its innovative approach and application in practice, as well as the system of work divided into sections.

5) Research method - no justification why e.g. the SWOT tool was not used to determine the most important researched postulates and (why the current method was chosen and the analyzes were limited to 2021 - the adopted criteria, limitations, data, description should be legible for a person who is not an expert in the field topic) The current description is not a sufficient tool to verify a possible hypothesis and achieve the stated goals. The authors should familiarize themselves with the literature on the subject and the methods of verification of individual factors, in particular, research should be conducted in such a way that it answers the questions posed in the article and not only can be verified, but also provides a field for discussion with other researchers.

6) In the results section, the quality of the presentation of test results is significantly limited. It would be prudent to use more than one research tool - defining its parameter - what the authors want to test or prove. The data presented in both the figures and tables are imprecise - and the division does not cover the essence of technological, economic or other solutions - and has no scientific value.

7) In the discussion chapter, the authors did not indicate and did not provide, for example, in a tabular form, what they understand as trends and prospects in the scope of the described drives, charging station technology, or economic factors affecting the development of electromobility.

  8) An important problem is the lack of a proposal for an own model based on the analysis of the literature on the subject - indication of directions of development or even directions of further research, considering that we are in 2023.

9) The Conclusions chapter should provide and focus on the prospects and directions of development for the future. Currently, it is an attempt to present considerations in the form of insufficiently described Scenarios. In the case of presenting prospects for the future, e.g. a formula, an algorithm would be required, in the case of recalling future research trends, e.g. based on an appropriate statistical method - therefore the scientific value in a foreign form is significantly limited.

  10) Literature - is too poor and outdated for the research problem described so extensively in writing, publications on individual issues in the area of the most important problems related to TCO, residual value, innovation in vehicle charging,increase in energy costs etc. issued under the MDPI should be cited.

Reviewer 2 Report

Clear and well presented work.

Author Response

Many thanks!

Reviewer 3 Report

This study analyzes scientific publications in Scopus relating to vehicle electrification and associated key enabling technologies. The topic is scientifically applicable and interesting. However, there are some concerns about the content. Specific comments and problems on the manuscript are provided below:

1.      This study analyzed the distribution of vehicle electrification (XEV) publications. However, it is recommended to summarize the research content of these publications. Because these are equally interesting and not just the number of publications.

2.      What is the difference in content and perspective between the vehicle electrification publications and the enabling technologies publications? And how many parts of the enabling technologies publication refer to vehicle electrification?

3.      It is recommended to obtain some interesting information from the perspective of these publication keywords.

4.      It is curious to know whether charging piles and power grids are also enabling technologies and whether their related research involves vehicle electrification.

5.      In the discussion or conclusion section, it is recommended to provide some outlook or strategic suggestions based on the results of the study.

Round 2

Reviewer 1 Report

After proofreading, the reviewer does not make any comments to the text.
I congratulate the authors on their idea and wish them creative continuation of research in this field.

Author Response

Many thanks for your help!

Reviewer 3 Report

Please add more latest refences and polish your language.

Author Response

Many thanks. We have now carried out a language check and added some references of relevance, some of them published after we started the work with this publication.